# Mining autonomous student patterns score on LMS within online higher education

Ricardo Ordoñez-Avila[1,2], Jaime Meza[1] and Sebastian Ventura[2]

[1] Departamento de Sistemas Computacionales, Universidad Técnica de Manabí, Portoviejo, Manabí, Ecuador

[2] Departamento de Informática y Análisis Numérico, Universidad de Córdoba, Córdoba, Córdoba, Spain



Corresponding authors
Ricardo Ordoñez-Avila,
ermenson.ordonez@utm.edu.ec
Sebastian Ventura, sventura@uco.es

## ABSTRACT

Higher education institutions actively integrate information and communication technologies through learning management systems (LMS), which are crucial for online education. This study used data mining techniques to predict the autonomous scores of students in the online Law and Psychology programs at the Technical University of Manabi. The process involved data integration and selection of more than 16,000 records, preprocessing, transformation with RobustScaler, predictive modelling that included recursive feature elimination with cross-validation to select features (RFEcv), and hyperparameter fitting to achieve the best fit, and finally, evaluation of the models using metrics of root mean square error (RMSE), mean absolute error (MAE), and the coefficient of determination ($R^2$). The feature selection framework suggested by RFEcv contributed to the performance of the models. The variables analyzed focused on download rate, homework submission rate, test performance rate, median daily accesses, median days of access per month, observation of comments on teacher-reviewed assignments, length of final exam, and not requiring the supplemental exam. Hyperparameter adjustment improved the performance of the models after applying RFEcv. The models evaluated showed minimal differences in RMSE ([0.5411 .. 0.6025]). The gradient boosting model achieved the best performance of $R^2$ = 0.6693, MAE = 0.4041 and RMSE = 0.5411 with the Law online program data, as with the Psychology online program data, with an $R^2$ = 0.6418, MAE = 0.4232 and RMSE = 0.6025, while the combination of both data sets reflected the best performance with the extreme gradient boosting (XGBoost) model with the values of $R^2$ = 0.6294, MAE = 0.4295 and RMSE = 0.5985. Future research and implementations could include autonomous score data through plugins and reports integrated into LMSs. This approach may provide indicators of interest for understanding and improving online learning from a personalized, real-time perspective.

# INTRODUCTION

Higher education institutions constantly strive to promote the use and management of information and communication technologies in online education (*Hatamleh & Hatamleh, 2024*; *Khajuria, Sharma & Sharma, 2023*). Learning management systems

(LMS) are widely used in this mode of study due to their continuous improvement, diverse functionalities, and ability to record data (*Al-Hunaiyyan, Al-Sharhan & AlHajri, 2020*; *Furqon et al., 2023*; *Cavus, Mohammed & Yakubu, 2021*). LMS has facilitated more autonomous education; however, in this mode, it is more complicated for teachers to establish precise assessments of their students' autonomous learning without face-to-face interaction (*Belkina, Lazorak & Yaroslavova, 2021*; *Osman, 2022*). Thus, without the ability to closely monitor assignments and daily progress, it is difficult to determine whether students are genuinely accessing course materials or simply completing tasks without assimilating essential knowledge.

LMS can provide important information about student activity and the sequence of their autonomous learning (*Ismail et al., 2021*). Autonomous learning in the virtual classroom is evidenced by the student's ability to manage their educational process, accessing a variety of resources that allow them to learn at their own pace and according to their individual needs (*Lee, 2016*; *Alharbi, 2022*; *Leshchenko, Lavrysh & Kononets, 2021*). Analyzing the records of interaction in the virtual classroom allows for identifying patterns that could predict academic performance (*Mehnen et al., 2023*). For instance, the frequency of access to the virtual classroom may be associated with higher grades (*Broadbent, 2016*).

Educational data mining (EDM) analyses data collected during teaching and learning by applying predictive techniques to student data. Prominent methods in the literature include analyzing student activities to predict their final grades based on performance (*Gaftandzhieva et al., 2022*). Machine learning algorithms such as random forests, support vector machines, logistic regression, naïve Bayes, and the k-nearest neighbor algorithm have been employed for these tasks (*Yağcı, 2022*). Furthermore, EDM has been used to extract data from the Moodle LMS log, identifying student patterns that can support autonomous learning and provide crucial information for teachers in implementing future pedagogical strategies (*Villegas-Ch & Luján-Mora, 2017*; *Abdullah, 2021*; *Araka et al., 2019*; *Tamada, Giusti & De Magalhaes Netto, 2021*; *Okike & Mogorosi, 2020*; *Bayazit, Apaydin & Gonullu, 2022*; *Hussain & Dimililer, 2021*; *Cabral & Figueira, 2019*). However, there are few studies focused on predicting students' autonomous scores based on their interactions on an LMS platform.

This study explores educational data mining models for predicting student autonomous scores based on interaction patterns in LMSs within online higher education. Techniques using non-parametric and parametric data are employed to achieve the best fit for different variables. By predicting student autonomous scores through analyzing their interaction in the virtual classroom, an approximation of their academic performance in this mode of study can be provided. This approximation can facilitate the early intervention of teaching improvement mechanisms upon detecting potential course success or failure. Therefore, this study delves into a recent field of research and could significantly contribute to the understanding and enhancement of online learning in higher education.

The following research questions were posed to guide this study: (RQ1) What is the relationship between students' interactions with the LMS and their levels of autonomous learning? (RQ2) Is there a relationship between the frequency of access to the virtual

classroom and academic success? (RQ3) What data mining techniques provide the highest accuracy in predicting student autonomy scores? (RQ4) How do student behaviors in the LMS affect their academic success?

The methods employed, the results of the prediction models, the discussion of the research questions, the limitations detected during this study, and the main conclusions organize the structure of this work.

## RELATED WORK

In educational data mining, classification techniques such as naive Bayes, J48, and support vector machines are commonly used; clustering algorithms like K-means and association rule mining, such as *A priori*, are also employed to predict academic performance (*Dol & Jawandhiya, 2024*). Additionally, algorithms with fuzzy methods have been used to explore and link student performance with teaching (*Mansouri, ZareRavasan & Ashrafi, 2021*; *Ordoñez-Avila et al., 2023*).

Several studies have explored how interactions within LMS can predict student performance. These investigations utilize LMS data to analyze student behavior and its correlation with academic performance (*Mehnen et al., 2023*; *Villegas-Ch & Luján-Mora, 2017*; *Okike & Mogorosi, 2020*; *Villegas-Ch et al., 2018*; *Cherniltsev & Panteleeva, 2021*; *Khan et al., 2023*; *Rienties, Toetenel & Bryan, 2015*). LMS data explored include quizzes, midterm and final exams, and the final course grade (*Wang, 2021*). Additionally, data on interaction through daily accesses are examined (*Fayaza & Ahangama, 2024*) and aspects related to academic performance, assignments, and social components (*Maraza-Quispe et al., 2022*). The feature analysis, such as the frequency of daily or monthly access, the use of classroom resources and other interactions in the LMS, can influence students' progress, impacting their learning outcomes and success in the course.

Regarding feature selection applied in educational data mining, some studies report using recursive feature elimination (RFE) to predict the career development of graduating students (*Li & Zhao, 2025*). Other approaches, such as school risk analysis, random forest and principal component analysis (PCA), were used as feature selection methods (*Ferdousee & Haque, 2024*). RFE was also employed in analyzing at-risk higher-education students, standing out for its results against the genetic algorithms technique (*Alwarthan, Aslam & Khan, 2022*). PCA has been the most widely used method for predicting student performance (*Dhankhar & Solanki, 2022*; *Ambadas et al., 2023*; *Li & Li, 2024*). Likewise, a variant of the Least Absolute Shrinkage and Selection Operator (LASSO) method, called Log-LassoNet, has been employed in combination with deep learning to interpret the prediction of student academic performance (*Luo & Wang, 2024*). Although selecting important features has been widely used in various approaches, there is little information on its use for specific purposes, such as analyzing students' autonomous scores.

Various machine learning algorithms have been used, including logistic regression, decision trees, naive Bayes, multilayer perceptron (MLP) neural network, support vector machines, random forests, sequential minimal optimization, and multiple linear regression, to predict student performance using LMS data (*Yacoub et al., 2022*; *Alhassan, Zafar & Mueen, 2020*; *Ramaswami et al., 2020*; *Nithiyanandam et al., 2022*; *Jović*

*et al., 2022*; *Xiong & Wu, 2024*). Random Forest (RF), XGBoost, k-nearest neighbor (KNN), and support vector machine (SVM), have also been used with 10-fold cross-validation for LMS data predictions, achieving performance close to 80% (*Gaftandzhieva et al., 2022*). Another model implementation with Random Forest has improved performance with objective features at 70% (*Hernández-García et al., 2024*). Tree-based models, such as extreme gradient boosting (XGBoost) and RF are more robust than linear regression due to their ability to handle nonlinear relationships, high dimensionality, and outliers. They capture complex interactions between variables without requiring strict assumptions about the data distribution (*Elith, Leathwick & Hastie, 2008*).

Prediction using regression algorithms has been explored in this context (*Xiong & Wu, 2024*; *Sridhara, Falkner & Atapattu, 2023*). *Arifin et al. (2023)* compared six regression algorithms to assess their effectiveness. The study combined academic and non-academic characteristics to predict student performance. The gradient-boosted regression tree showed the lowest error rate among the models examined. To measure error, several metrics such as R-squared, adjusted R-squared, mean absolute error (MAE), mean squared error (MSE), and the root of MSE (RMSE) have been used to compare the models (*Wang, 2021*; *Spiess & Neumeyer, 2010*).

The role of machine learning models in information extraction in educational data mining has facilitated the analysis of complex data obtained from LMS. Models such as RF have been used to identify at-risk students (*Fahd, Miah & Ahmed, 2021*); techniques such as KNN, RF and SVM, have been applied in the analysis of learning styles (*Maulany, Santosa & Hidayah, 2024a*, *2024b*; *Nguyen, 2023*). Likewise, k-means and SVM have been employed to explore features that influence academic performance (*Ahmed & Al-Omari, 2024*). The gradient boosting regressor has been used to predict students' cumulative grade point average (*Aslam et al., 2024*), and the multiple logistic regression model has been helpful in the study of student retention and dropout (*Bertolini, Finch & Nehm, 2023*).

Other studies have explored autonomous learning and interactions in virtual classrooms. For example, *López-Goyez, González-Briones & Chamorro (2025)*, proposes designing architectures in the Moodle platform for the academic monitoring of students in an online modality, using autonomous agents that analyze the interaction between the student and the virtual tutor. In turn, *Tsai (2021)*, examines the analysis of questionnaires to evaluate autonomous learning strategies, student behavior in the virtual classroom and their interaction with the materials. In addition, it identifies significant correlations between online learning activities and the autonomy perceived by the student, suggesting that specific interaction patterns may influence the development of their academic independence.

## MATERIALS AND METHODS

This study uses Moodle as the LMS to mining student interactions in an online higher education environment. Advanced data analysis techniques were employed to extract, select, and preprocess relevant information from student activities. The analyses included

normality tests, correlation analyses, and normalization techniques to adjust the data for non-linear predictive models.

## Raw data

Student interaction and scores records from a Moodle Learning Management System (LMS) used in an online higher education environment were analyzed. The data originated from the Faculty of Humanities and Social Sciences, specifically from 42 virtual classrooms of the Online Law Program and 40 virtual classrooms of the online Psychology program of the Technical University of Manabí during the first academic term of 2023.

## Selection

For data extraction, Python 3.8 and libraries such as pandas, numpy, and functools were used to manipulate the data. Extensive data integration work was conducted, including combining queries, appending, and transposing data. Each log file from the LMS and the grades were downloaded from the virtual classroom, with each course coded. Each log file contained over 200,000 records.

Student grades were read from an Excel file from the LMS, selecting relevant columns such as the final student score, partial teaching grades (C1), autonomous work (C2), and practical and experimental learning (C3), in addition to the exam grade, the identifier code of each subject, and the student identifier. Subsequently, C1 (30%), C2 (20%), and C3 (20%) were combined into a single variable named autonomous_score, representing 70% of the total course grade (100%), excluding the exam (30%). This step allowed the creation of the first data frame (DF1). These weights correspond to the instructional design of courses for each component, C1 through C5, for 100 points per course.

Logs or records from the LMS were read using the student and course identifier. Features focused on the submission of assignments and quizzes were extracted, including both the quantity and the performance rate of each; the duration in minutes of final and supplementary exams; feedback provided by the teacher after reviewing an assignment, viewed by the student; the number of content downloads per unit, as well as the overall download rate; the average and median of daily accesses and the days accessed per month were also obtained. These data contributed to creating a second data frame (DF2). These fields are selected based on increased performance as a function of interactions. Moderate and strong relationships were preliminarily tested between the fields obtained from Moodle logs and the ratings.

All data frames were merged with unique student and course fields. To process the Moodle records, separate data frames were needed to compute new columns. Figure 1 shows the data integration process, specifying the data frames produced and the columns extracted to compile the final dataset. It also describes the procedures applied using SQL and Python, reducing data loss by controlling the data separately. Finally, three data sets were generated for post-testing: one for each program of study and one combining both programs.

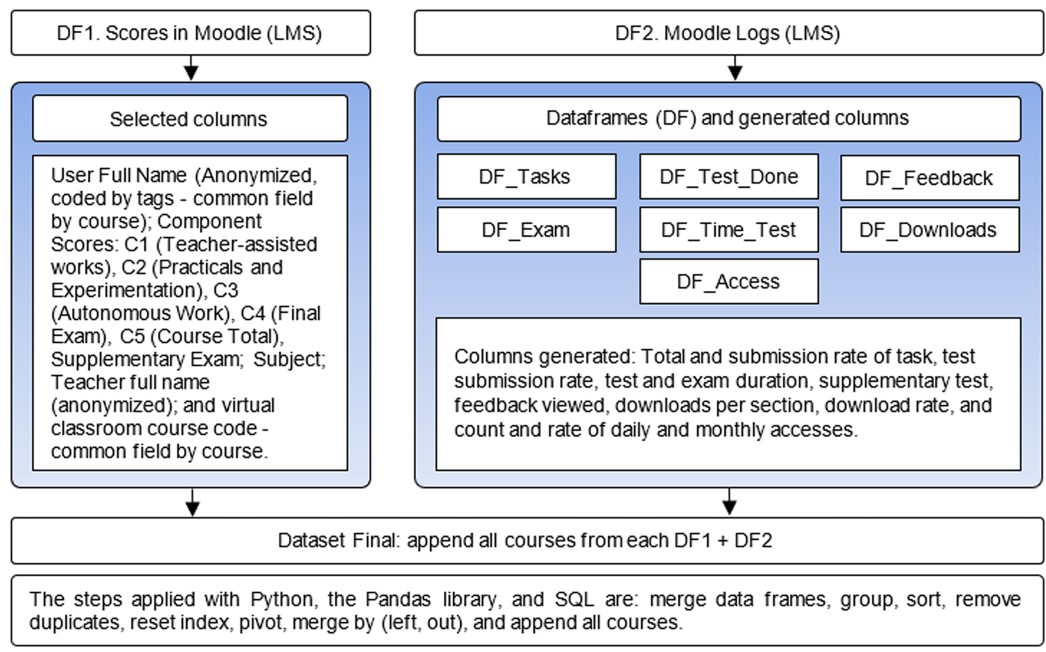

**Figure 1** **Data integration.** How scores and interaction data from LMS files can be merged using Python and SQL.      

## Preprocessing

After loading and assembling the final dataset, comprising grades and Moodle LMS records, preprocessing steps were carried out. First, column headings were renamed, checked for missing or null data, evaluated for the presence of outliers, and explored for measures of central tendency. Subsequently, normality tests were performed using the Shapiro-Wilk, Kolmogorov-Smirnov, and Anderson-Darling methods.

In addition, Spearman correlation analysis was performed for non-normally distributed data. From the correlation coefficients obtained, a set of predictor variables was selected based on their demonstration of moderate to strong relationships and their relevance to the study object, guided by domain knowledge. A multicollinearity analysis was then performed to test for possible correlations between predictor variables. Multicollinearity was validated using the variance inflation factor (VIF) method, where a VIF coefficient more significant than 10 indicates high multicollinearity.

## Transformation

Transforming integer values to numeric values was necessary to prepare for analysis in non-linear regression prediction models. Normalization and scaling tests were conducted using StandardScaler, MinMaxScaler, and RobustScaler methods. StandardScaler normalizes features with a mean of zero and a standard deviation of one. MinMaxScaler transforms features by scaling them to a range from 0 to 1. Conversely, RobustScaler adjusts features using statistics resistant to outliers by removing the median and scaling the data according to the interquartile range. These methods were evaluated to determine which most effectively enhances the results of the final model.

## Predictive models

This analysis used a computer equipped with an integrated Intel® Iris Xe Graphics GPU, an additional NVIDIA GeForce MX450 GPU, 16 gigabytes of RAM, and an eleventh-generation Intel Core i7 processor. It is advisable to utilize optimal processing and memory resources due to the computational effort required by model optimization methods, primarily. The response variable (autonomous_score) (C1_C2_C3) was defined based on the correlation between the combination of the first three grading components and the final course grade. The data partitioning assigned 75% to the training set and the remaining 25% to the test set. This configuration was applied to three data sets analyzed: the law program, psychology, and a combination of both.

Nonlinear models were employed for prediction using advanced regression algorithms. Unlike traditional regression models, such as linear and logistic, which assume linear relationships, these nonlinear models can handle complex behavior and the presence of outliers, overcoming the limitations that affect the accuracy of linear models. A preliminary performance evaluation of candidate nonlinear algorithms was performed for this study. They were considered in the previous analysis, Decision Tree, SVM regressor (RBF Kernel), neural network (multi-layer perceptron, MLP), gradient boosting, XGBoost, and RF to address nonlinearity and increase robustness in the presence of outliers in the analysis (*Shao, Ahmad & Javed, 2024*).

To choose the most relevant variables, we explored the performance of recursive feature elimination methods with cross-validation (RFEcv), principal component analysis (PCA) and the least absolute selection and shrinkage operator with cross-validation (LASSOcv). The Recursive Feature Elimination with Cross-Validation (RFEcv) technique, employing the Python scikit-learn library, was used to optimize model performance. Several sets of candidate variables were initially analyzed using ten folds for cross-validation. This process culminated in selecting the set that provided the best balance between model complexity and predictive capacity. The results are expanded in the next section.

Hyperparameter optimization was also applied using grid search techniques in conjunction with cross-validation. Grid search systematically evaluates combinations of hyperparameters, testing each set on different data folds to enforce model robustness through cross-validation. This comprehensive method allows for selecting the best hyperparameter configuration to maximize model performance by controlling overfitting.

## Evaluation

To evaluate the performance of the models, the metrics of root mean square error (RMSE), mean absolute error (MAE), and the coefficient of determination $R^2$ were used. RMSE (Eq. (1)) measures the difference between values predicted by the model and the observed values, calculated as the square root of the average of the squares of the differences between predictions and actual values. RMSE helps evaluate the accuracy of prediction models and is sensitive to outliers. It provides a measure in the same units as the dependent variable.

$$RMSE = \sqrt{\frac{1}{n} \sum_{i=1}^{n} (y_i - \hat{y}_i)^2} \qquad (1)$$

where $\hat{y}_i$ are the values predicted by the model, $y_i$ represents the actual values, and $n$ is the number of observations.

MAE (Eq. (2)) was used to measure the magnitude of the error between predictions and observed values. MAE measures the error in the same units as the dependent variable and provides an intuitive interpretation of model performance. However, MAE does not differentiate between small and large errors as effectively as RMSE. Therefore, a combination of metrics is often required to evaluate model performance.

$$MAE = \frac{1}{n} \sum_{i=1}^{n} |y_i - \hat{y}_i| \qquad (2)$$

where $\hat{y}_i$ are the predicted values, $y_i$ are the actual values, and $n$ is the total number of observations.

The coefficient of determination $R^2$ (Eq. (3)) was used to indicate the proportion of the variability in the dependent variable that the model explains. A value close to one indicates a firm fit, although it does not necessarily imply good generalization power. In nonlinear models, this metric should be interpreted with other indicators. $R^2$ is defined as:

$$R^2 = 1 - \frac{\sum_{i=1}^{n} (y_i - \hat{y}_i)^2}{\sum_{i=1}^{n} (y_i - \bar{y})^2} \qquad (3)$$

where $\hat{y}_i$ represents the values predicted by the model, $y_i$ are the actual values, $\bar{y}$ are the mean of the observed values, and $n$ are the total number of observations.

## RESULTS

In total, about 16,900 instances were generated in the data integration. Each dataset contains 34 candidate columns generated during the selection phase. Table 1 lists 15 preliminary columns chosen for correlation analysis.

The data are integers and numeric, reporting patterns of complex, non-linear behaviors. No missing values were detected, but there were outliers in the response variable. Given the nature of the data's behavior, no methods were applied to manipulate the outliers in this work. The extreme scores range from 0 to 40. It is most common for students to prepare for a particular possibility of success from 40 points upwards. Minority cases can be detected for the potential implementation of actions within the framework of academic management to promote improvements in performance in actual practice.

Several statistical tests were conducted to determine whether the data follow a normal distribution. The results of these tests indicate an apparent deviation from normality. The null hypothesis H0, that the data follow a normal distribution, and the alternative hypothesis H1, that the data do not follow a normal distribution, were defined.

The Shapiro-Wilk test, known for its potential and sensitivity in samples of various sizes (*Arnastauskaitė, Ruzgas & Bražėnas, 2021*), was conducted. With Shapiro-Wilk, an extremely low *p*-value was obtained (See Table 2), indicating a significant deviation from

**Table 1 Columns produced in the data integration.** Each row shows the column name, data type, and description.

| Id | Column | Type | Description |
|----|--------|------|-------------|
| 1 | C1 | int64 | Grade for the "teaching" component, synchronous and asynchronous activities. |
| 2 | C2 | int64 | The grade for the "autonomous work" component and synchronous and asynchronous activities. |
| 3 | C3 | int64 | Grade for the "practical and experimental learning" component and synchronous and asynchronous activities. |
| 4 | autonomous_score | int64 | Sum of C1, C2, and C3. Later, the response variable is used in predictive modelling. |
| 5 | total_task_submitted | int64 | A total number of assignments submitted. |
| 6 | task_submission_rate | float64 | Proportion between assignments submitted and offered. |
| 7 | total_test_submitted | int64 | Total number of quizzes submitted. |
| 8 | test_submission_rate | float64 | Proportion between quizzes submitted and offered. |
| 9 | median_test_duration | float64 | Median duration of quizzes in minutes. |
| 10 | no_supplementary_exam | int64 | Indicator (1 or 0) of whether an additional exam was required, with one meaning not required. |
| 11 | final_exam_duration | int64 | Duration of the final exam in minutes. |
| 12 | feedback_viewed | int64 | Indicator (1 or 0) of whether the student reviewed the teacher's feedback. |
| 13 | download_rate | float64 | The proportion of content downloaded relative to the minimum download per unit. Thresholds: 0.25, 0.50, 0.75, and 1 for four units. |
| 14 | median_daily_access | float64 | The median of daily accesses. |
| 15 | median_days_per_month | int64 | The median of accesses per number of days each month. |

**Table 2 Statistical tests for normal distribution.**

| Test | Statistic | p-value | Result |
|------|-----------|---------|--------|
| Shapiro-Wilk | 0.9569 | <0.001e−19 | Non-normal |
| Kolmogorov-Smirnov | 0.1036 | <0.001e−19 | Non-normal |
| Anderson-Darling | 77.984 | Critical values: 0.576, 0.656, 0.787, 0.918, 1.092 | Non-normal |

normality. Additionally, the Kolmogorov-Smirnov test, which contrasts the empirical distribution of the data with a theoretical normal, also rejected the hypothesis of normality. This technique was complemented by the Anderson-Darling test, which was particularly effective in detecting deviations in the distribution's tails. The convergence of these results confirms that the data do not fit a normal distribution, thus justifying the exploration of non-parametric methods or transformations in subsequent analyses.

Significantly low $p$-values in the normality tests strengthen the evidence against H0 in favor of H1. Therefore, insufficient statistical evidence exists to accept the null hypothesis (H0), as the data do not follow a normal distribution. The Spearman correlation, suggested for non-parametric data, was conducted. Moderate and low correlations were identified. The lower correlations also suggest that each variable might provide unique information to the model, essential for understanding different student behaviors or performance aspects. The multicollinearity analysis was validated through the VIF, with coefficients ranging from 1.01 to 1.99, indicating a shallow relationship among the predictors. Therefore, there is no concerning multicollinearity among most of the explanatory variables in the model.

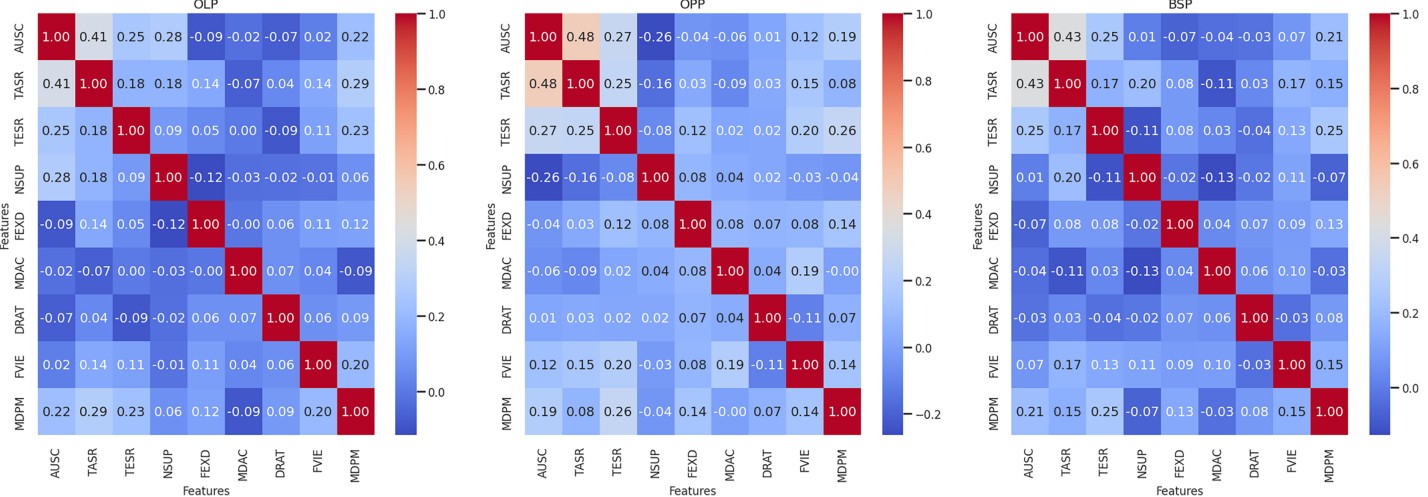

**Figure 2 Spearman correlation in the data sets: OLP, online Law program; OPP, online Psychology program; and BSP, both study programs.** The Spearman correlation coefficients between all candidate variables for data modelling. Correlations are shown for each data set corresponding to the OLP, OPP and BSP. The variables analyzed were: autonomous_score (AUSC), task_submission_rate (TASR), test_submission_rate (TESR), no_supplementary_exam (NSUP), final_exam_duration (FEXD), median_daily_access (MDAC), download_rate (DRAT), feedback_viewed (FVIE), and median_days_per_month (MDPM).

In addition, the Robust Scaler was the best data scaling method compared to the MinMax Scaler and Standard Scaler.

## Predictive models

Interactions with different models allowed us to determine the most effective one for the treated data source, for which the Spearman correlation coefficient was 0.80 between the response variable and the final course grade. These three components suggest the student's autonomous activity and are related to quizzes, assignments, practical activities, and other frequently dedicated efforts, excluding the final exam. They represent 70 points out of a total of 100 in the course.

Figure 2 show the correlation coefficients of the variables analyzed for the OLP, OPP, and BSP data sets, respectively. Spearman's correlation matrix reveals positive and moderate correlations between autonomous score and homework submission rate, with values of 0.41, 0.48, and 0.43 for each OLP, OPP, and BSP dataset, respectively. To a lesser degree, the variables that correlate with the autonomous score are the median number of days accessed per month and the rate of tests sent, not taking the supplementary exam, with correlation coefficients between 0.19 and 0.28 in all data sets. Other variables such as final exam duration, median number of accesses per day, download rate, and observed feedback show very low correlations with coefficients between 0.01 and 0.12.

Table 3 shows three sections of results from testing the candidate models with three different data sets, the first corresponding to data from the online Law program, the second with data from the Psychology program, and the third block with the union of both curricula in a single data set. After identifying the appropriate variables, it was decided to

**Table 3 Preliminary tests of candidate models for each study program data set: online Law program (OLP), online Psychology program (OPP), and both study programs (BSP).**

| Model | OPP | | | OLP | | | BSP | | |
|---|---|---|---|---|---|---|---|---|---|
| | MAE | RMSE | $R^2$ | MAE | RMSE | $R^2$ | MAE | RMSE | $R^2$ |
| Decision tree | 0.5345 | 0.7807 | 0.3438 | 0.5430 | 0.8155 | 0.3371 | 0.5665 | 0.8373 | 0.2712 |
| Random forest | 0.4287 | 0.5865 | 0.6297 | 0.4474 | 0.6517 | 0.5766 | 0.4527 | 0.6428 | 0.5705 |
| Gradient boosting | 0.4136 | 0.5551 | 0.6683 | 0.4298 | 0.6088 | 0.6306 | 0.4580 | 0.6424 | 0.5709 |
| XGBoost | 0.4201 | 0.5715 | 0.6484 | 0.4366 | 0.6380 | 0.5943 | 0.4445 | 0.6293 | 0.5883 |
| SVR | 0.4222 | 0.6032 | 0.6082 | 0.4173 | 0.6510 | 0.5775 | 0.4260 | 0.6409 | 0.5730 |
| Neural network | 0.4383 | 0.5909 | 0.6241 | 0.4432 | 0.6373 | 0.5951 | 0.4475 | 0.6377 | 0.5772 |

test these prediction models due to the nonlinear and complex behavior of the predictors. This step was performed to identify which models would be most suitable for further testing.

The results illustrated in Table 3 confirm the choice of RF, gradient boosting and XGBoost. These models achieved the best preliminary balance between the lowest error (MAE and RMSE) and highest coefficient of determination ($R^2$). The mean and standard deviation for the chosen models are presented below:

MAE = 0.4368 ± 0.0150
RMSE = 0.6140 ± 0.0352
$R^2$ = 0.6086 ± 0.0364

Next, the recursive feature elimination results with cross-validation, the estimation of hyperparameters, their application, and the evaluation of the prediction models are presented.

### Recursive feature elimination with cross-validation

PCA and LASSOcv showed lower performance in the selected models, with RMSE values of 0.7 and 0.6 and $R^2$ coefficients of 0.4 and 0.5, respectively, compared to the higher values of RFEcv. Tests were performed using eight to five manually selected predictor variables to identify the features recommended for removal and those of the highest relevance in each model and dataset. Recursive feature elimination was performed independently for each dataset to infer differential behaviors between the two study programs and their combination.

RFE progressively eliminated less relevant features, offering higher interpretability compared to PCA. Although cross-validation optimizes model performance, its application involves higher computational costs, requiring repeated training with different subsets of features. On the other hand, PCA can be more efficient in reducing dimensionality by mathematical decomposition (*Casuat & Festijo, 2020*), especially on large data volumes. However, since this study prioritizes model interpretability and optimization, RFE has proven more efficient in this context.

**Table 4 Performance indicators of recursive feature elimination (RFEcv) for the OLP dataset.** Each row shows the tests with the model used, the features removed, the number of features selected (N features) and their selector score, the 10-fold cross-validation score (cross_val_score), and the prediction performance values $R^2$, MAE and RMSE. (*) The best fit occurs with six predictors using the gradient boosting model.

| Model | Features removed | N features (selector score) | cross_val_score (standard deviation) | $R^2$ | MAE | RMSE |
|---|---|---|---|---|---|---|
| Gradient boosting | None | 8 [0.6928] | 0.67 [0.03] | 0.6599 | 0.4137 | 0.5517 |
| XGBoost | None | 8 [0.8388] | 0.66 [0.03] | 0.6266 | 0.4204 | 0.5781 |
| Random forest | None | 8 [0.9375] | 0.64 [0.03] | 0.6140 | 0.4308 | 0.5878 |
| Gradient boosting | Feedback_viewed | 7 [0.6918] | 0.67 [0.03] | 0.6617 | 0.4122 | 0.5503 |
| XGBoost | Median_daily_access | 7 [0.8090] | 0.66 [0.03] | 0.6448 | 0.4133 | 0.5638 |
| Random forest | Feedback_viewed | 7 [0.9353] | 0.64 [0.03] | 0.6110 | 0.4317 | 0.5901 |
| *Gradient boosting | Median_daily_access, feedback_viewed | 6 [0.6924] | 0.68 [0.03] | 0.6670 | 0.4094 | 0.5459 |
| XGBoost | Median_daily_access, feedback_viewed | 6 [0.8099] | 0.66 [0.03] | 0.6387 | 0.4163 | 0.5687 |
| Random forest | Download_rate, feedback_viewed | 6 [0.9305] | 0.63 [0.03] | 0.5994 | 0.4379 | 0.5988 |
| Gradient boosting | Median_daily_access, download_rate, feedback_viewed | 5 [0.6896] | 0.67 [0.03] | 0.6661 | 0.4083 | 0.5467 |
| XGBoost | Median_daily_access, feedback_viewed, median_days_per_month | 5 [0.7716] | 0.65 [0.04] | 0.6464 | 0.4112 | 0.5626 |
| Random forest | No_supplementary_exam, download_rate, feedback_viewed | 5 [0.9205] | 0.60 [0.03] | 0.5607 | 0.4558 | 0.6270 |

Table 4 presents the results of the online Law program. The RFEcv recommends an optimal number of between seven and eight variables in this data set. However, testing indicated that the Gradient Boosting model performed best with six selected predictors, achieving RMSE = 0.5459, MAE = 0.4094 and $R^2$ = 0.6670. In this iteration, RFEcv suggests the removal of median daily accesses and seen feedback, as neither variable is relevant for this dataset. In addition, if only five predictors were used, this method recommends excluding the download rate and the variable associated with not taking the supplemental exam.

Figure 3 shows the importance of the features in the three datasets and the three best-performing models. Overall, the most relevant feature is the task submission rate, followed by the test submission rate, failure to take the supplemental exam, final exam duration, and median monthly access days. In the OLP and OPP sets, with gradient boosting, the median number of daily accesses, download rate, and feedback viewed presented low significance in predicting the response variable, indicating a limited impact. However, in the BSP set, with XGBoost, the feedback variable ranked among the four most relevant variables, suggesting differences in the perception of importance according to the model and the dataset's structure.

Table 5 presents the results of feature selection for the online Psychology program dataset. The gradient boosting model obtained the best performance with seven variables, compared to the six variables selected for the online Law program. In both cases, the median number of accesses per day is recommended to be removed, suggesting its low relevance in predicting the response variable. The performance coefficients obtained were

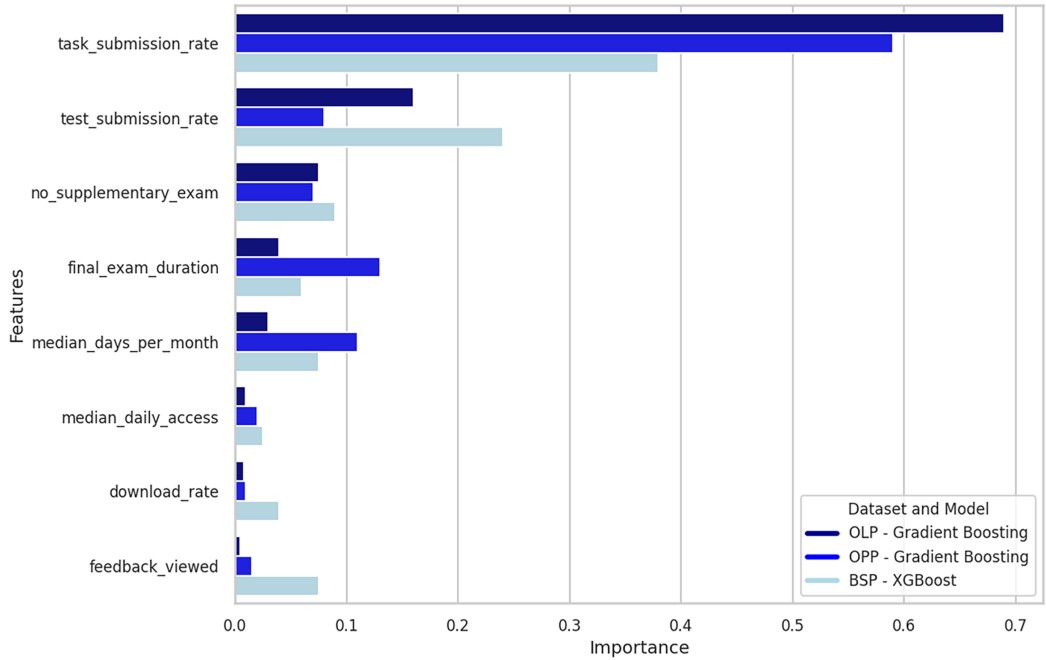

**Figure 3 Importance of characteristics in RFEcv testing and best-performing models.** The importance levels of the features obtained in the tests with the gradient boosting and XGBoost models. Gradient boosting presented the best performance with the online Law program (OLP) and online Psychology program (OPP) datasets. At the same time, the XGBoost model showed the best performance with the combined set of both study programs (BSP).

**Table 5 Performance indicators of recursive feature elimination (RFEcv) for the OPP dataset.** Each row shows the tests with the model used, the features removed, the number of features selected (N features) and their selector score, the 10-fold cross-validation score (cross_val_score), and the prediction performance values $R^2$, MAE and RMSE. (*) The best fit occurs with seven predictors using the XGBoost model.

| Model | Features removed | N features (selector score) | cross_val_score (standard deviation) | $R^2$ | MAE | RMSE |
|---|---|---|---|---|---|---|
| Gradient boosting | None | 8 [0.6650] | 0.63 [0.03] | 0.6336 | 0.4325 | 0.6092 |
| XGBoost | None | 8 [0.8553] | 0.57 [0.04] | 0.6044 | 0.4319 | 0.6331 |
| Random forest | None | 8 [0.9386] | 0.59 [0.04] | 0.6065 | 0.4387 | 0.6314 |
| *Gradient boosting | median_daily_access | 7 [0.6669] | 0.63 [0.03] | 0.6330 | 0.4294 | 0.6073 |
| XGBoost | median_daily_access | 7 [0.8271] | 0.57 [0.03] | 0.5886 | 0.4397 | 0.6456 |
| Random forest | download_rate | 7 [0.9352] | 0.58 [0.04] | 0.5997 | 0.4432 | 0.6369 |
| Gradient boosting | median_daily_access, download_rate | 6 [0.6626] | 0.62 [0.03] | 0.6306 | 0.4331 | 0.6117 |
| XGBoost | median_daily_access, download_rate | 6 [0.8168] | 0.57 [0.03] | 0.5945 | 0.4396 | 0.6409 |
| Random forest | download_rate, feedback_viewed | 6 [0.9307] | 0.57 [0.04] | 0.5865 | 0.4512 | 0.6473 |
| Gradient boosting | median_daily_access, download_rate, feedback_viewed | 5 [0.6549] | 0.62 [0.03] | 0.6209 | 0.4393 | 0.6198 |
| XGBoost | median_daily_access, feedback_viewed, median_days_per_month | 5 [0.7519] | 0.52 [0.04] | 0.5699 | 0.4470 | 0.6601 |
| Random forest | no_supplementary_exam, download_rate, feedback_viewed | 5 [0.9216] | 0.52 [0.04] | 0.5490 | 0.4705 | 0.6760 |

**Table 6 Performance Indicators of Recursive Feature Elimination (RFEcv) for the BSP dataset, both study programs.** Each row shows the tests with the model used, the features removed, the number of features selected (N features) and their selector score, the 10-fold cross-validation score (cross_val_score), and the prediction performance values $R^2$, MAE and RMSE. (*) The best fit occurs with seven predictors using the XGBoost model.

| Model | Features removed | N features (selector score) | cross_val_score (standard deviation) | $R^2$ | MAE | RMSE |
|---|---|---|---|---|---|---|
| Gradient boosting | None | 8 [0.6067] | 0.59 [0.01] | 0.5815 | 0.4556 | 0.6360 |
| XGBoost | None | 8 [0.7726] | 0.59 [0.01] | 0.6098 | 0.4363 | 0.6141 |
| Random forest | None | 8 [0.9312] | 0.57 [0.02] | 0.5893 | 0.4486 | 0.6301 |
| Gradient boosting | median_daily_access | 7 [0.6081] | 0.59 [0.01] | 0.5818 | 0.4549 | 0.6358 |
| *XGBoost | median_daily_access | 7 [0.7563] | 0.60 [0.01] | 0.6135 | 0.4299 | 0.6112 |
| Random forest | feedback_viewed | 7 [0.9352] | 0.56 [0.02] | 0.5776 | 0.4546 | 0.6390 |
| Gradient boosting | median_daily_access, download_rate | 6 [0.6074] | 0.59 [0.01] | 0.5806 | 0.4547 | 0.6367 |
| XGBoost | median_daily_access, download_rate | 6 [0.7512] | 0.59 [0.01] | 0.6071 | 0.4341 | 0.6163 |
| Random forest | download_rate, feedback_viewed | 6 [0.9211] | 0.55 [0.02] | 0.5566 | 0.4636 | 0.6547 |
| Gradient boosting | median_daily_access, download_rate, feedback_viewed | 5 [0.6001] | 0.59 [0.01] | 0.5777 | 0.4587 | 0.6390 |
| XGBoost | final_exam_duration, median_daily_access, download_rate | 5 [0.6696] | 0.57 [0.02] | 0.5650 | 0.4537 | 0.6484 |
| Random forest | no_supplementary_exam, download_rate, feedback_viewed | 5 [0.9042] | 0.52 [0.02] | 0.5219 | 0.4823 | 0.6798 |

RMSE = 0.6073, MAE = 0.4294 and $R^2$ = 0.6330, consolidating gradient boosting as the technique with the best performance in estimating students' autonomy levels in the separately analyzed programs.

In contrast, integrating both data sets suggests that the best-performing model was XGBoost, reaching RMSE = 0.6112, MAE = 0.4299 and $R^2$ = 0.6135. In this configuration, seven variables were used, eliminating the median number of daily accesses and reaffirming its low relevance in predicting autonomy levels (see Table 6). These results provide a reference on the potential generalizability of the model, which could be relevant for future studies incorporating new data from other careers and approaches, particularly in STEM fields.

The selection of characteristics reflected variations among the different cohorts of students analyzed in this study. For the Law online program, six of eight variables were selected, eliminating the median number of daily accesses and comments viewed. In contrast, only the median number of daily accesses was removed in the Psychology online program and in the data set for both degrees. In the tests performed for all three datasets, RFE suggested removing the variable "median number of daily accesses" in the best-performing models, gradient boosting and XGBoost. However, only gradient boosting was recommended to remove the observed feedback variable in a study program.

Since the median daily accesses were removed in all feature selection tests using RFE, its high dispersion and variability in student behavior in online education settings might justify its exclusion in future experiments, especially in different educational contexts. These results provide a comparative benchmark on the variability of selected characteristics in different datasets, cohorts, or curricula. Nevertheless, RFE may identify

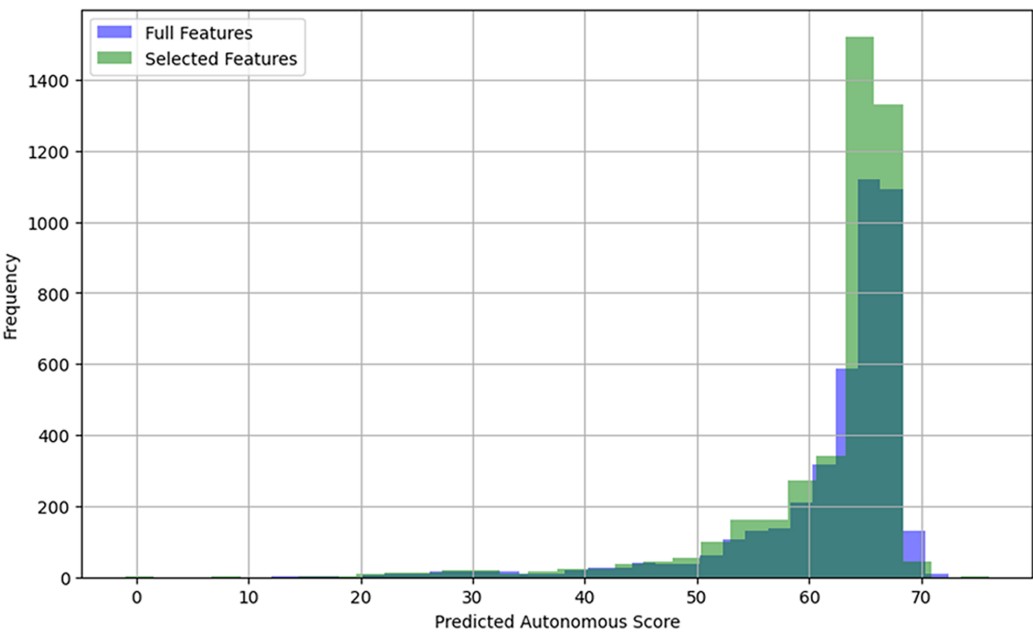

**Figure 4  Autonomous score distribution with RFE and XGBoost.**

common patterns, depending on the instructional and methodological design of the course within the same academic context, suggesting a potential generalization of the model.

Figure 4 shows the distribution of the autonomous score data using the XGBoost model on the combined dataset of both analyzed careers. It also presents a comparative histogram showing the distribution of the target variable before and after feature selection with RFEcv.

The superposition of the histograms allows for the visualization of how the models perceive the influence of the features on the prediction of the autonomous score. A more concentrated distribution after the RFEcv application suggests that the selected features contribute less noise, which could improve model stability compared to using all available variables.

### Model tuning with hyperparameters

Using hyperparameters improved the RMSE of the tests performed on the data. Table 7 details the parameters applied in the fits, with the results for each data set, the eliminated variables, and the performance in training and testing to provide a reference of the final fit of each model. Finally, after recursive feature selection, the application of hyperparameter fitting gradually improved the performance indicators $R^2$, MAE and RMSE.

### Model evaluation

The models were evaluated using the RMSE metric. The data were split into training and test sets using a 75 to 25 ratio, respectively, with random_state = 42. Initially, tests performed without model optimization showed evidence of overfitting, with an adjusted

**Table 7 Model performance with hyperparameter optimization programs.** Each row presents the tests performed for each dataset, with the best-performing model and features removed. The R$^2$, MAE, and RMSE tests also include the best hyperparameters and the R$^2$ and standard deviation (sd) training performance coefficients.

| Dataset | Model | Features removed | **best_params (10-folds) | R$^2$ [sd] train | R$^2$ test | MAE | RMSE |
|---|---|---|---|---|---|---|---|
| OLP | Gradient boosting | median_daily_access, feedback_viewed | learning_rate: 0.05<br>max_depth: 5<br>min_samples_leaf: 4<br>min_samples_split: 10<br>n_estimators: 200 | 0.6879 [0.03] | 0.6693 | 0.4041 | 0.5411 |
| OPP | Gradient boosting | median_daily_access | learning_rate: 0.05<br>max_depth: 5<br>min_samples_leaf: 2<br>min_samples_split: 5<br>n_estimators: 100 | 0.6357 [0.03] | 0.6418 | 0.4232 | 0.6025 |
| BSP | XGBoost | median_daily_access | colsample_bytree: 0.8<br>earning_rate: 0.05<br>max_depth: 5<br>min_child_weight: 3<br>n_estimators: 200<br>subsample: 0.8 | 0.6294 [0.01] | 0.6294 | 0.4295 | 0.5985 |

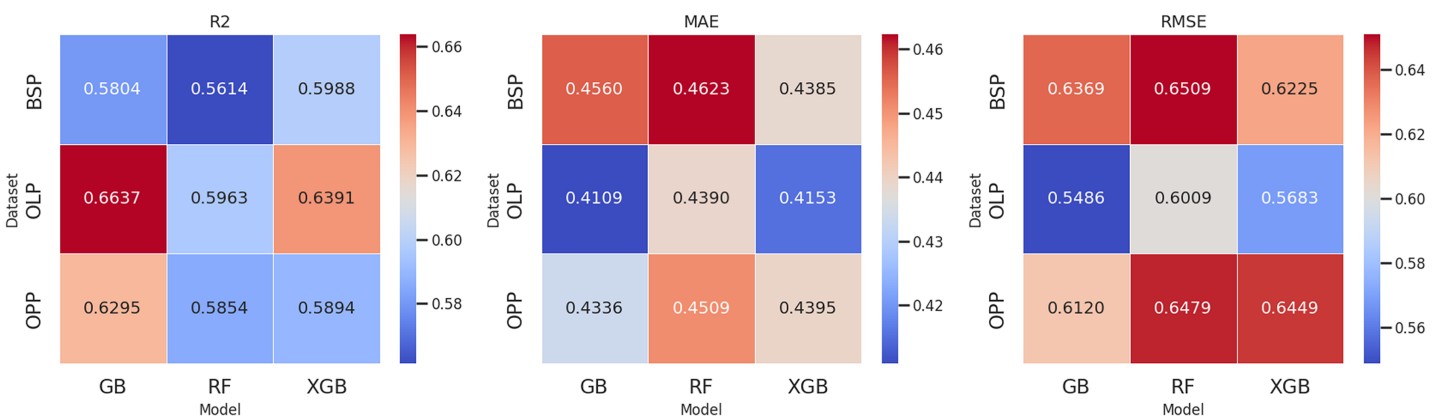

**Figure 5 Comparison of model performance by data set.**

R-squared of 0.99 in both training and test sets and an MSE of 1.57e−07 in training and 0.0007 in the test set.

Figure 5 presents the evaluated models' mean performance values, considering each dataset's R$^2$, MAE, and RMSE metrics. These results correspond to the iterations performed during feature selection before hyperparameter fitting, providing a comparative view of the models' preliminary performance.

With hyperparameter fitting, the best performance for the online Law program data set was RMSE = 0.5411, while the online Psychology program achieved an RMSE = 0.6025.

For its part, the combination of both data sets resulted in an RMSE = 0.5985. In all three cases, the elimination of the variable "median number of accesses per day" is maintained in common, suggesting its low relevance in predicting student autonomy levels. In the second iteration, statistically, RFEcv in the XGBoost and gradient boosting model suggested eliminating the median daily access. Based on domain knowledge, virtual classroom styles, and the nature of the data, tests were conducted by eliminating up to three features.

In a comparative framework between the gradient boosting, XGBoost, and RF models, no statistically significant differences were found in any metrics evaluated ($R^2$, MAE, and RMSE). For this purpose, the normality of the values was verified using the Shapiro-Wilk test. The results confirmed that the data followed a normal distribution in all the metrics evaluated, which justified the use of ANOVA for the analysis of differences.

The p-values obtained for $R^2$ (0.3451), MAE (0.5547), and RMSE (0.4928) were more significant than 0.05, indicating that there is insufficient evidence to claim that any of the models have significantly superior predictive performance. Therefore, they can be considered equivalent.

Regarding computational efficiency, feature selection took between 1 and 2 min for all models. Hyperparameter search using Grid Search, gradient boosting, and RF took between 30 and 40 min, while XGBoost completed the task in less than 2 min, making it suitable for real-time monitoring applications in LMS.

## DISCUSSION

The autonomous score's outliers helped identify the LMS's interaction pattern, allowing us to predict certain cases correctly. No methods were applied to treat extreme values of the response variable. On the other hand, specific predictors were calculated using the median, which is more stable than the mean in the face of outliers. The choice to keep using extreme data in the autonomous score response variable and not to use resampling techniques is based on the intention to preserve the authenticity of the events in the context of online education, where minority cases can provide valuable information on real behaviors.

Synthetic data in this context could generate biases, reducing the ability to generalize in practical scenarios. We recognize the unbalanced nature of the data and rely on the ability of robust predictive models, which, by combining multiple weak tree models, iteratively improve the performance of the final model by adjusting for residual error. Finally, the model must recognize and correctly interpret these rare but significant behaviors.

To answer research question RQ1: What is the relationship between students' interactions with the LMS and their levels of autonomous learning? Spearman's correlation analysis was applied together with recursive feature elimination with cross-validation (RFEcv). These methods allowed us to identify the most relevant variables in determining the levels of autonomous learning. Five variables, the task submission rate, test submission rate, not to perform the complementary exam, median number of access days per month, and final exam duration, better represented the importance of features in the three data sets analyzed.

On the other hand, the observed feedback, download rate, and median number of accesses per day variables were systematically suggested for elimination by RFEcv in different iterations of the models. This suggests that the behavior of these variables could be influenced by the course's instructional design, teaching strategy, and the nature of the online study modality.

To answer research question RQ2: Is there a relationship between the frequency of access to the virtual classroom and academic success? Spearman's correlation analysis was applied. The results revealed moderate coefficients between the median number of monthly access days and the autonomous score, with values ranging from 0.19 to 0.22. However, the total number of accesses presented considerable variability over time, with very low correlation coefficients, between 0.01 and 0.06. In this context, applying methods such as time series analysis to evaluate the daily frequency of accesses could generate noisy results due to the possible volatility of the series and the absence of a clear pattern.

On the other hand, the median number of days of access per month showed greater stability, evidencing a more consistent monthly behavior pattern regarding access for reviewing or downloading content, submitting assignments, and taking exams.

Three nonlinear tree-based regression models were evaluated to answer research question RQ3: What data mining techniques provide the highest accuracy in predicting student autonomy scores? Using non-linear models such as RF, gradient boosting, and XGBoost allowed for exploring non-parametric data, seeking the best fit with model optimization techniques for predicting students' autonomous scores. RF, gradient boosting, and XGBoost were chosen for their ability to handle nonlinear data, resist outliers, and avoid overfitting, especially in educational contexts with complex data from platforms such as Moodle.

The RF model improves prediction in complex data contexts using multiple decision trees built from bootstrap samples and randomly selected feature subsets. This approach offers excellent resistance to outliers and allows capturing non-linear relationships without prior transformations, leveraging the aggregation of multiple trees to stabilize predictions (*Shao, Ahmad & Javed, 2024*; *Breiman, 2001*). Gradient boosting and XGBoost are ensemble techniques that build trees sequentially, learning from the errors of previous trees to minimize a loss function. XGBoost adds regularization (L1 and L2), which helps prevent overfitting and is effective in high-dimensional contexts. Both methods are suited to handle non-linearity and provide robustness against outlier data (*Sarker, 2021*).

The results indicated that gradient boosting and XGBoost achieved the best performance in model fitting. In addition, 10-fold cross-validation was applied, which proved to be more computationally efficient than Leave-One-Out CV, which represents a higher computational cost due to the volume of data analyzed. Also, using recursive feature elimination with cross-validation (RFEcv) allowed a more accurate selection of predictor variables compared to other methods, such as PCA and LASSOcv. Additionally, the search and adjustment of hyperparameters contributed to the optimization of the final model, showing minimal variations in the coefficients of $R^2$, MAE, and RMSE. These techniques improved the accuracy of predicting students' autonomy scores.

To answer the question RQ4: How do student behaviors in the LMS affect their academic success? The relationship between predictors and the response variable was analyzed and complemented with the selection of characteristics. The results suggest that more significant interaction in the classroom may be associated with higher correlation coefficients and greater importance of certain variables in the predictive models. In this sense, some of the variables identified in this study are relevant, suggesting their potential generalizability and applicability in future research. On the other hand, certain variables could be further tested to evaluate their impact in different educational environments and teaching modalities and validate their influence on students' academic success.

The suggested features in these models represent objective data. However, their interpretation of the autonomous score could be subject to the subjectivity and nature of the context or case study. Furthermore, instructional design influences might emerge. Knowing the different ways of organizing the virtual classroom, its resources, the course pedagogy, and the learning design from activities and the virtual classroom influence how students participate online and impact learning performance (*Rienties, Toetenel & Bryan, 2015*). For example, the median of daily accesses shows considerable instability among participants. Although it does not correlate strongly with the response variable, it could be relevant in other contexts and research. This result would be based on the observed relationship between continuous daily effort and overall course success outcomes and, thus, the autonomous score.

RMSE, $R^2$ and MAE were used to evaluate the models as metrics to compare performance and determine the best fit. Other metrics, such as adjusted $R^2$, were not considered since the main focus was on the direct interpretation of the error. Adjusted R-squared is a version of R-squared that adjusts for the number of predictors in the model, which helps avoid overfitting that could occur by adding additional variables to the model (*Karch, 2020*). However, its applicability in non-linear regressions needs to be investigated. For linear regression models, adjusted R-squared is a reliable metric. In contrast, it is not applicable in non-linear models due to how these models fit, mainly because they need to rely on ordinary least squares minimization to ensure this relationship. Due to these fundamental differences, many experts suggest other metrics, such as MSE, Akaike information criterion (AIC), or Bayesian information criterion (BIC), to evaluate the goodness of fit in non-linear models (*Chicco, Warrens & Jurman, 2021*).

Finally, by applying models tested in educational data mining techniques, the prediction of students' autonomous scores based on their interaction patterns in online higher education LMS was explored; therefore, this study allowed the analyzing variables that exhibited complex patterns and non-normal data through non-linear models and optimization methods in a field with limited prior research.

## Limitations

Extending more data from more than two curricula, in addition to different contexts or institutions, instructional designs, and fields such as technical and medical sciences, could offer greater generalizability. However, the tests conducted in this study offer an alternative

separate modelling approach that could be useful for exploring autonomous scoring with the suggested predictors.

In addition, new predictors related to the instructional design of each institution and the regulatory methods of online study modalities in other fields of science, technology and engineering could be incorporated. One possible predictor could be interaction in forums. In this study, information on forums was not included due to reduced or no interaction in virtual classrooms related to the nature of the instructional design of the curriculum or the institution. Standard demographics, such as gender and geographic location, could be added to future research to explore further results. However, this study did not consider them due to the considerable variation in cases or their reduced significance in classroom interactions.

The need for balanced samples due to the imbalance in the numeric values of the response variable represented a significant limitation. Although resampling techniques such as Synthetic Minority Oversampling Technique (SMOTE) (*Alrumaidhi, Farag & Rakha, 2023*) have been used in classification work and have shown promising results, this study decided to exclude imputation methods to preserve the originality of the data. The data naturally present extreme values in the program analysis, where outliers are less than 40 points of the autonomous score, constituting a minority. In contrast, the most common values range between 40 and 70, corresponding to students who pass the course.

Due to the nature of the gradient boosting and the XGBoost model, the function learned can be very complex and cannot be easily expressed in a closed formula. Therefore, this work does not present a closed formula for prediction. However, the proportion of importance of each variable can be estimated as a weight in an approximation to explain the autonomous score.

## CONCLUSIONS AND FUTURE WORK

This research proposes a set of indicators for online education, such as material download rates, assignment and exam submission, median daily access, monthly access days, review of comments on assessed assignments, final exam duration, and skipping of the supplemental exam. Future research can explore these indicators to understand and improve learning in this educational modality.

Non-linear models were tested, achieving the best fit by eliminating the predictor median_daily_access. The median of daily access should be an essential measure in future tests; however, for this dataset, the access pattern is irregular per day. On the other hand, a better trend was found in the median of accesses per day per month, with this predictor being a more balanced data point to explain the autonomous score.

The optimal tuning of the models, based on the selection of predictors, was influenced by the nature of the data and the configuration style of the virtual classroom. In an open configuration environment, where the marking of completed tasks and the mandatory download of materials are not required, it is essential to consider these factors to adequately reflect student behavior and performance. This flexibility in the learning environment implies that predictive models must adapt to the lack of structured data and the diverse ways students interact with the content.

Using recursive feature elimination methods with cross-validation helped find objective features validated in the prediction models. Parameter estimation, in turn, facilitated the detection of the best-performing model when combined with the results of both optimization methods. The gradient boosting and XGBoost models showed the best performance; however, it is important to consider computational trade-offs, as tree-based models require more training time compared to simpler regression models.

The variables included in this study represent an initial proposal of predictors that can be tested in other environments with other data. It is essential to consider the nature of each course and the particularities of each field of study, which may change in pedagogy, design, and configurations in the virtual classroom during each academic period. The importance of using the median instead of the mean in some of the created features lies in improving correlation results between predictors and the response variable. Additionally, due to the presence of outliers and skewed distributions in the data set, it was decided to describe the central tendency of the data.

Among the variables chosen for analysis, demographic and socioeconomic data of the students are not included. This work explores data strictly linked to the virtual classroom and the student's interaction with the course. It is not particularly interesting to recognize the student's gender, for example. All features associated with interaction in the LMS provide an approximation to the autonomous score from the perspective of the student's dedication and autonomy in progressing in their studies. It is essential to apply methods that ensure data security and privacy. This study employed coding labels for courses and students as column identifiers in the data integration processes. The processed data set does not include sensitive student information.

This work proposed a roadmap that could be replicated for each nature of data and implications related to classroom design. Its structure modifications could significantly influence how data are collected and analyzed. Future research directions could test the model in different online learning environments, expand the dataset, or integrate deep learning techniques. Exploring hybrid approaches, such as combining feature selection with deep learning models, could improve predictive accuracy, favoring scalability and real-world implementation in future work.

In the analyzed data, students are not required to download material from the virtual classroom. However, future designs could include this requirement, giving greater weight to the download rate variable. Therefore, its exploration is suggested in future works. Additionally, more active integration of forums and other interactive resources could offer additional student participation and engagement indicators, thus facilitating a deeper analysis of their behavior and performance. These potential changes reflect the evolution of pedagogical practices and enrich the database with new dimensions to explore in future research.

Future research could test the integration of machine learning models into LMSs. Moodle plugins may help deploy reports and analytics frameworks that provide important measures of real-time autonomous learning. This practice can contribute to academic management and timely teacher intervention to improve personalized insights and overall course outcomes.

Therefore, this work provides a reference framework for new studies, presenting the potential of these predictive models in analyzing complex behaviors of students' autonomous performance in higher education through the information in Moodle LMS logs.

## ACKNOWLEDGEMENTS

We would like to thank the University of Cordoba, Spain, in particular Prof. Sebastian Ventura as leader of the Knowledge Discovery and Intelligent Systems (KDIS) group, for his scientific contribution, and his guidance and supervision of the development of the doctoral thesis entitled "Data mining model for the prediction of teacher evaluation based on student performance".

### Funding

The authors received no funding for this work. The APC was supported by MICIN/AEI/10.13039/501100011033 under Grant PID2023-148396NB-I00. The funders had no role in study design, data collection and analysis, decision to publish, or preparation of the manuscript.

### Grant Disclosures

The following grant information was disclosed by the authors:
APC: MICIN/AEI/10.13039/501100011033, PID2023-148396NB-I00.

### Competing Interests

Sebastián Ventura is an Academic Editor for PeerJ.

### Author Contributions

- Ricardo Ordoñez-Avila conceived and designed the experiments, performed the experiments, analyzed the data, performed the computation work, prepared figures and/or tables, authored or reviewed drafts of the article, and approved the final draft.
- Jaime Meza conceived and designed the experiments, performed the experiments, analyzed the data, performed the computation work, prepared figures and/or tables, authored or reviewed drafts of the article, and approved the final draft.
- Sebastian Ventura conceived and designed the experiments, performed the experiments, analyzed the data, performed the computation work, prepared figures and/or tables, authored or reviewed drafts of the article, and approved the final draft.

### Data Availability

The numerical grade data and calculated fields created for the response variable and predictors from the LMS records of an online course of study and code are available in the Supplemental Files.

## Supplemental Information

Supplemental information for this article can be found online at http://dx.doi.org/10.7717/peerj-cs.2855#supplemental-information.

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
