# Peer review of "Mining autonomous student patterns score on LMS within online higher education"

_PeerJ Computer Science, doi:10.7717/peerj-cs.2855_

## Round 0.1 · original submission · Minor Revisions

Dear Authors,

Thank you for submitting your article. Reviewers have now commented on your article and suggest revisions. Although extensive, they are all minor comments. We do encourage you to address the concerns and criticisms of the reviewers and resubmit your article once you have updated it accordingly. Furthermore, explanations of the variables of Equation 1 should be correctly written in italic. Figure 1 should be corrected. Figure 2 and Figure 4 should be polished.

Best wishes,

·

Basic reporting

The manuscript is clearly written, using professional language. It outlines the role of Learning Management Systems (LMS) in online higher education and how mining data from LMS logs can provide insights into students' autonomous learning. The background and context of the study are well-developed, and the literature review comprehensively covers relevant research in Educational Data Mining (EDM).

Suggested Improvements:

The literature review can be enhanced by adding more recent studies, particularly focusing on the role of machine learning models in educational data mining and LMS interactions

The figures and tables presented in the study are helpful, but their captions could benefit from more detailed explanations to improve clarity.

Experimental design

The experimental design is robust, utilizing data from Moodle LMS logs, along with non-linear predictive models such as Gradient Boosting, Random Forest, and XGBoost, to predict students' autonomous scores. The methodology, including data preprocessing, feature selection, and hyperparameter tuning, is clearly explained and logically structured.

Suggested Improvements:

Provide more details about the specific preprocessing steps applied to handle outliers and missing data.
The hyperparameter tuning process is briefly mentioned, but more information about how hyperparameters were chosen and optimized would help improve the replicability of the results.

Validity of the findings

The results are valid and backed by the experimental analysis. The Gradient Boosting model performed the best with an RMSE of 0.5469, which indicates good predictive capability for students' autonomous scores. The study's emphasis on using non-linear models to capture complex behaviors in student interactions is well-justified.

Suggested Improvements:

Discuss the potential limitations of the current model, including whether the dataset is sufficient for generalizing the results across different courses or educational institutions.
Include more future directions, such as how this model can be expanded or tested on other datasets or how it could incorporate additional variables such as student demographics.

Additional comments

The manuscript presents a valuable contribution to the field of educational data mining by exploring the use of machine learning models to predict student performance in online education. The use of recursive feature elimination and hyperparameter tuning enhances the model's predictive performance.

General Comments:

Consider expanding the discussion on the practical implications of this study, such as how the models can be integrated into LMS systems for real-time monitoring and intervention strategies.
Including a brief discussion of the ethical considerations related to student data privacy would add more depth to the paper, particularly given the sensitive nature of educational data.
Thank you for the opportunity to review this manuscript. The work is relevant and well-structured, and the suggestions provided aim to enhance the quality and impact of the research.

·

Basic reporting

The manuscript is written in clear, professional English. This component is satisfactory.

The manuscript provides adequate references and context in the field of educational data mining and online learning. More detailed justification on the selection of specific models (Gradient Boosting, Random Forest, XGBoost) over others could strengthen the case.
The manuscript would benefit from a section that explains the selection of non-linear models, emphasizing the specific reasons for their robustness in handling the complexities of LMS data. For example, the authors should discuss why the characteristics of Random Forest and Gradient Boosting (e.g., handling non-linearity and overfitting) make them better suited for this dataset.

The article follows an organized structure, including an introduction, methods, results, and conclusion sections. The figures and tables are relevant and high-quality, helping explain the modeling process and outcomes. This component is satisfactory.

The authors have shared their raw data and describe their use of it in various preprocessing stages, meeting data transparency requirements. However, further clarification on data imbalances, particularly extreme values, could improve transparency.

The results are well-formatted, with the necessary definitions provided for the models and metrics. Clear mathematical explanations for techniques like RMSE are present, though the results section could benefit from additional clarity on the implications of feature selection outcomes.

Experimental design

The research aligns with the journal's aims, as it addresses the problem of predicting autonomous student scores using LMS data. The focus on autonomous scores is a novel approach compared to traditional student performance metrics.

The manuscript clearly defines its research question: predicting autonomous scores based on LMS data. The identified knowledge gap, particularly concerning autonomous learning behaviors in virtual classrooms, is meaningful.

The investigation follows a methodologically rigorous approach, using multiple predictive models. However, the lack of exploration into why some features (e.g., download rates, daily access) perform poorly raises questions about the deeper reasons behind the findings. This component is important for readers who seek to apply the results in their practice.

For example, the authors could hypothesize why “feedback viewed” or “download rate” were less impactful in this study. Was it because of the specific course structure, the quality of the content, or student behavior patterns? A deeper dive into these findings would make the results more interpretable for readers unfamiliar with the specific LMS being used.

Methods are described in sufficient detail, allowing future researchers to replicate the study. The use of Python libraries and specific algorithms is clearly outlined.

Validity of the findings

The manuscript provides sufficient underlying data, and the methods used for handling non-normal distributions (e.g., Spearman correlation) are appropriate. However, the rationale behind excluding resampling techniques (like SMOTE) could be further expanded to explain why imbalanced data was retained. Data originality is important, but relying solely on unbalanced data can lead to biased model performance, particularly when predicting minority outcomes. Detail justifications should be provided for this methodological choice.

The conclusions are well-linked to the original research question. The study provides insights into autonomous learning behaviors, although the authors could better address how these findings can be generalized to other LMS environments or educational contexts.
The authors could improve by discussing the broader applicability of their findings. How would these models perform if applied to STEM fields or courses with different pedagogical structures? A more explicit discussion on the generalizability of these findings to other programs or institutions would strengthen the manuscript’s impact.

Additional comments

Ethical Considerations and Data Privacy Issue.

The manuscript touches briefly on ethical standards but does not deeply engage with the potential ethical concerns related to mining student data from LMS systems.

Adding a dedicated section or paragraph on ethical considerations, including how student privacy was ensured during data collection and analysis, would be valuable. This is especially important given the increasing scrutiny around data privacy in educational research. The authors could discuss anonymization techniques or the ethical use of predictive modeling in educational settings, particularly how it might impact student behavior or interventions.

Reviewer 3 ·

Basic reporting

Authors should carefully address each of the following comments and concerns (Given in Additional comments), giving thoughtful consideration to any suggested improvements. If certain comments cannot be fully addressed, these should be acknowledged as limitations within the study’s limitations section. Clearly identifying these areas will enhance the transparency and rigor of the research.

Experimental design

Authors should carefully address each of the following comments and concerns (Given in Additional comments), giving thoughtful consideration to any suggested improvements. If certain comments cannot be fully addressed, these should be acknowledged as limitations within the study’s limitations section. Clearly identifying these areas will enhance the transparency and rigor of the research.

Validity of the findings

Authors should carefully address each of the following comments and concerns (Given in Additional comments), giving thoughtful consideration to any suggested improvements. If certain comments cannot be fully addressed, these should be acknowledged as limitations within the study’s limitations section. Clearly identifying these areas will enhance the transparency and rigor of the research.

Additional comments

Introduction discusses the importance of LMS in the context of online higher education and the role of Educational Data Mining (EDM) in predicting students' autonomous scores. However, there are certain gaps and areas for improvement:
1. Literature Review: A more comprehensive reference to existing studies would strengthen the context of this research, particularly by providing a detailed overview of studies examining the relationship between autonomous learning and LMS interactions.
2. Research Problem and Aim: Clearly articulating the primary research problem and aim would help readers better understand the study's significance and motivation. For instance, it should be made more explicit why predicting students' autonomous scores is essential in this research.
3. Methodological Approaches: Adding more details on the methodological approaches would be beneficial. Providing specific information about the data mining techniques used would allow readers to grasp the methodology of the study more effectively.
4. Interaction Measures: Clarifying the criteria for evaluating student interactions would enhance the study’s rigor. Detailing which variables will be used in interaction analyses and how they relate to academic success would strengthen the research framework.

Including sample research questions in the introduction can more clearly outline the purpose and scope of the study.
To further delineate the study's objectives, the following research questions are proposed:
1. What is the relationship between students' LMS interactions and their levels of autonomous learning?
2. Which methods are most effective in predicting autonomous scores based on LMS interaction models?
3. Is there a connection between the frequency of access to the virtual classroom and academic success?
4. How do autonomous learning strategies influence LMS interactions, and what is their effect on student engagement?
5. Which data mining techniques provide the highest accuracy in predicting student autonomous scores?
6. What impact do student behaviors, such as forum participation and material downloads, have on their academic success?
These questions will guide the research and clarify the study's scope, thereby enhancing its methodological approach and overall contribution to understanding and improving online learning in higher education.


Comments on dataset, method, and analysis:

1) Data Source Diversity and Representativeness: The interaction data extracted from the LMS pertains only to one faculty (Faculty of Humanities and Social Sciences) and a single program (Online Law Program), which limits the generalizability of the study. Incorporating data from different departments or programs could allow for a broader assessment of student interactions. This limitation should be clearly stated.
2) Detailing Features: It would be beneficial to provide a more detailed explanation of the obtained features. For instance, the relationship between features such as assignment submission frequency or the amount of content downloaded and student success should be clarified. More information should be provided regarding the selection criteria for these features and their significance for the model.
3) Limitations in the Data Integration Process: Potential issues encountered during the data integration process have not been mentioned. The large volume of LMS log files (with over 200,000 records in each file) may pose challenges in accurately merging and transferring the data comprehensively. The measures taken during this process (such as maintaining data quality and preventing data loss) and the challenges faced should be explained.
4) Determining Weights in Feature Selection: It is stated that features such as C1, C2, and C3 have a 70% impact on determining student grades; however, the criteria for this weighting have not been discussed. The process of establishing these weights, their relationship with other features, and their impact on the model should be elaborated.
5) Potential Information Loss from Feature Merging: Following the merging of the DF1 and DF2 data frames, no analysis has been conducted on the combined meaning of the features. It should be evaluated whether there is potential information loss resulting from this merging process and how this may affect the model's performance.
6) Inadequacy of Data Preprocessing Steps: The data preprocessing process can be elaborated further. For instance, although steps related to missing values are mentioned, the specific methods used for imputation (e.g., mean, median, or regression methods) are not clearly stated. A more systematic explanation of the missing data strategy is crucial for the replicability of the study.
7) Alternative Approaches to Data Normalization: The study exclusively utilized StandardScaler, but other scaling methods suitable for different models (e.g., MinMaxScaler, RobustScaler) should also be evaluated. This can help achieve more stable results, particularly with data that have different distributions.
8) Limited Feature Selection Methods: Only Recursive Feature Elimination (RFE) was employed for feature selection, whereas other feature selection methods (e.g., PCA, LASSO) could also be tested. Comparing various methods may provide better insights into which features yield optimal model performance.
9) Model Validation Techniques: The model validation process in the study was conducted solely through k-fold cross-validation; however, the model may also need to be tested with different data splitting strategies. Particularly, adding more model validation techniques (e.g., Leave-One-Out CV) could strengthen the assessment of the model's generalizability.
10) Diversification of Evaluation Metrics: The RMSE metric alone may not be sufficient. To comprehensively evaluate model performance, additional evaluation metrics such as R² score or Mean Absolute Error (MAE) should be employed. This would help gain a better understanding of the model's overall accuracy and predictive performance.
11) Handling Outliers: The study mentions that the data contains outliers; however, it does not explain how these outliers were addressed. Dealing with outliers during the data cleaning phase before modeling can enhance the model's accuracy and stability.
12) Insufficiency in Model Selection: Only three models (Random Forest, Gradient Boosting, XGBoost) were utilized. It is recommended to subject different non-linear models (e.g., Support Vector Regression, Neural Networks) to testing in order to compare model performance. This could provide more detailed insights into which model delivers the best performance.
13) Necessity of the R2: R² allows for a contextual understanding of how much variation in the dependent variable is explained by the independent variables. This can be particularly useful in fields such as social sciences, where understanding the influence of various factors on outcomes is critical. R² serves as a straightforward statistic that can be communicated to stakeholders, providing a clear indication of model performance. This is particularly valuable in applied settings, where non-technical audiences may need to understand the effectiveness of a model in predicting outcomes. In summary, R² is a vital statistic for assessing the explanatory power of regression models. Its ability to quantify variance, facilitate model comparison, and guide improvements makes it an essential tool in statistical analysis and machine learning applications. However, it is important to use R² in conjunction with other metrics and analyses to gain a comprehensive understanding of model performance.
14) Inadequacy of Time Series Analysis: Trends noted regarding the use of daily or monthly access rates can be supported by a more in-depth time series analysis. Since students’ online learning behaviors may change periodically, this analysis can contribute to a better understanding of trends and increase the accuracy of the model.
15) Limited Activity Indicators: The study has only examined a few specific variables (such as material downloads, assignment submissions, and access durations) and suggests the integration of more interactive resources, such as forums, for future research. The absence of these resources in the current study may provide a limited perspective on students' levels of engagement in online education.
16) General spelling rules: The small paragraphs in the study make it difficult to read. They should be combined. Each section should have fewer paragraphs.

Addressing these shortcomings will enhance the methodological rigor of the study, provide a valuable and replicable research framework for a broader audience, and strengthen the generalizability and validity of the findings.

Cite this review as

---

## Round 0.2 · Minor Revisions

Dear Authors,

Thank you for addressing the reviewers' comments. Notwithstanding the fact that two of the reviewers have accepted your paper, Reviewer 1 believes that it requires minor revisions.

Best wishes,

·

Basic reporting

I have reviewed the manuscript "Mining Autonomous Student Patterns Score on LMS within Online Higher Education," which presents an investigation into the predictive modeling of student autonomy scores using Learning Management System (LMS) interaction data. The study applies data mining techniques, feature selection, and hyperparameter optimization to predict student performance within online higher education settings. The authors explore the role of Recursive Feature Elimination with Cross-Validation (RFEcv), Gradient Boosting, and XGBoost models in improving predictive accuracy using data from 16,000+ student records. This study is valuable in advancing educational data mining (EDM) by identifying key predictors of autonomous learning. Below, I provide constructive feedback to enhance the manuscript’s clarity, technical rigor, and impact.

Content and Structure
The manuscript is technically sound and well-organized, but specific sections require further elaboration to improve clarity and completeness.

Abstract:
The abstract effectively summarizes the study, but it should:

Explicitly highlight the novelty of the work, particularly how the proposed feature selection framework (RFEcv) improves the accuracy of predictive models.
Include comparative performance results of the Gradient Boosting, XGBoost, and Random Forest models, quantifying improvements in RMSE, MAE, and R².
Discuss real-world implications, such as how educational institutions can apply the model for real-time student monitoring and personalized learning.
Introduction:
The introduction provides a strong motivation for using EDM techniques in online learning, but it could be further improved by:

Clarifying the research gap, particularly how existing EDM studies have not adequately explored feature selection techniques for predicting student autonomy scores.
Discussing the impact of LMS interaction variables, such as how the frequency of access to learning materials correlates with student success.
Providing a stronger justification for model selection, explaining why tree-based models (XGBoost, Random Forest) outperform linear regression techniques in this context.
Conclusion:
The conclusion summarizes key findings effectively, but should:

Discuss practical deployment considerations, such as how the model could be integrated into LMS platforms like Moodle for real-time student monitoring.
Address computational trade-offs, particularly whether tree-based models introduce increased training time compared to simpler regression models.
Propose concrete future research directions, such as testing the model on different online learning environments, expanding the dataset, or integrating deep learning techniques.

Experimental design

Technical Clarifications and Suggestions
Feature Selection and Predictive Modeling
The manuscript introduces Recursive Feature Elimination with Cross-Validation (RFEcv) to enhance predictive accuracy, but should clarify:
How feature selection impacts model generalizability across different student cohorts.
The computational trade-offs of using RFEcv compared to simpler feature selection methods like Principal Component Analysis (PCA).
Why certain LMS features (e.g., median daily accesses) were excluded during feature selection.
Comparative Analysis of Machine Learning Models
The manuscript compares multiple predictive models, but should:
Include statistical significance testing (e.g., Wilcoxon signed-rank test, ANOVA) to confirm whether the performance differences between XGBoost and Random Forest are statistically meaningful.
Analyze feature importance variations, particularly how different machine learning models rank LMS interaction features.
Assess computational efficiency, providing execution time comparisons for different models.
Scalability and Real-World Deployment
The manuscript briefly discusses real-world applicability, but should:
Quantify computational overhead, particularly whether XGBoost’s training time is feasible for real-time LMS monitoring.
Assess generalizability, considering how the model performs across different academic disciplines.
Explore hybrid approaches, such as combining feature selection with deep learning models for improved predictive accuracy.
Visualization and Analysis
The manuscript includes useful figures and tables, but additional visualizations would improve clarity:

Feature Importance Analysis: Provide bar plots highlighting the most influential LMS features in predicting student autonomy.
Performance Comparison of Models: Use a heatmap comparing RMSE, MAE, and R² across different models and datasets.
Class Distribution Analysis: Include histograms showing the distribution of student autonomy scores before and after feature selection.

Validity of the findings

Literature Review and Citations
The manuscript presents a well-researched literature review, but additional references should be incorporated to strengthen the discussion on EDM and feature selection.

·

Basic reporting

The authors have satisfactorily expanded the manuscript to provide a clearer justification for the selection of non-linear models of choice in handling LMS data.

Further clarification was added regarding data imbalances and outliers, and the presentation of feature selection results has been improved. These additions enhance the clarity and completeness of the manuscript’s reporting.

Experimental design

The authors have addressed my concerns regarding the interpretation of less impactful features by discussing possible reasons and conceptual connections of such features. The methodological rigor of the study has been strengthened through these clarifications.

Validity of the findings

The authors have provided a well-reasoned explanation for not applying resampling techniques like SMOTE, emphasizing the importance of maintaining data originality in the context of rare but meaningful cases.

Furthermore, they have added discussion on the limitations of generalizing findings to other LMS environments and educational programs, recommending future research in varied fields such as STEM.

These enhancements improve the transparency and validity of the findings.

Additional comments

The manuscript now includes a paragraph addressing ethical considerations, specifically regarding data privacy, anonymization, and the importance of applying information security measures. This addition appropriately responds to concerns about the ethical use of LMS data in educational research.

Reviewer 3 ·

Basic reporting

Basic reporting has been improved with the arrangements made.

Experimental design

The corrections made to the methodology have enabled the study to mature.

Validity of the findings

I believe that the scope and verification of the study's findings are sufficient.

Additional comments

The authors addressed the comments thoroughly and implemented the necessary revisions effectively. Notably, the inclusion of well-defined research questions has enhanced the clarity of the study's purpose and justification. The consolidation of smaller paragraphs has improved the overall coherence and flow of the text. One of the most significant improvements was the addition of the R-squared metric, which allows for a clearer comparison between the models. Furthermore, the clarification of the study's limitations has provided valuable insights for future research. The inclusion of the MAE (Mean Absolute Error) metric was also a positive addition. At this stage, I believe that incorporating time series models is not essential, as the current methodology has yielded sufficient and robust findings. In its present form, I am confident that the study will make a meaningful contribution to the field.

Cite this review as

---

## Round 0.3 · accepted · Accept

Dear Authors,

Thank you for clearly addressing the reviewers's comments. Your paper now seems sufficiently improved and ready for publication.

Best wishes,

·

Basic reporting

This study uses data mining techniques to predict autonomous student scores in online higher education courses, specifically in Law and Psychology programs at the Technical University of Manabí. The authors used Learning Management System (LMS) data from Moodle, including over 16,000 records, to model and predict student performance in relation to autonomous learning behaviors. Various regression models, including Gradient Boosting and XGBoost, were applied and evaluated based on performance metrics such as RMSE, MAE, and R². The paper highlights key factors that affect autonomous learning, such as task submission rates, exam duration, and feedback engagement, offering new insights into online learning patterns.

Strengths:

The integration of multiple data sources and advanced data mining techniques is a significant contribution to understanding autonomous student learning patterns.

The use of various feature selection techniques (RFEcv, PCA, and LASSOcv) provides a robust approach to improving model performance and interpretability.

The study’s ability to predict autonomous learning scores provides actionable insights that could benefit both academic institutions and educators.

Significance of the Study:
This study is highly relevant in the context of online higher education, where understanding and improving autonomous learning is crucial for student success. By identifying patterns in LMS data and linking them to academic performance, the study has the potential to inform teaching strategies and interventions, particularly for enhancing student engagement and learning outcomes in virtual environments.

Suggestions for Improvement:

The paper could benefit from a clearer discussion of the practical applications of the predictive models, especially in terms of how they could be integrated into real-world online education systems for early intervention.

The authors could expand on the challenges and limitations of implementing such models in smaller institutions or resource-limited settings.

2. Content and Structure Review
Evaluating the Abstract, Introduction, and Conclusion:
The abstract effectively summarizes the goals, methodology, and key findings. However, it could provide more details on the specific metrics of model performance, such as the actual values of R², RMSE, and MAE, for a more thorough overview. The introduction provides a solid background on the role of LMS in autonomous learning but could benefit from a more in-depth review of similar studies to better position the current research in the field. The conclusion does a good job of summarizing the study’s contributions, but it could further explore the implications for future research and practical applications, particularly how the findings can be used to support personalized learning paths.

Suggestions for Improvement:

Include specific performance metrics (e.g., R², RMSE) in the abstract to provide a clearer summary of the study’s impact.

The introduction could be strengthened with additional references to recent studies and technologies in the area of educational data mining.

The conclusion should provide more concrete recommendations for real-world applications and future research directions.

3. Literature Review and Citation Updates
Ensuring Relevant References are Cited:
The literature review provides a comprehensive overview of key concepts and previous work in educational data mining (EDM). However, there is limited discussion on the role of hybrid models, combining multiple machine learning techniques for better prediction accuracy in online education environments. Additionally, more attention could be paid to the integration of artificial intelligence (AI) and machine learning in real-time learning analytics systems.

Suggestions for Additional Citations: To strengthen the literature review and broaden the context, I recommend incorporating the following recent references:

DOI: 10.1016/j.eswa.2023.122147
DOI: 10.54216/JAIM.090102
DOI: 10.54216/MOR.030205
DOI: 10.1007/s11540-024-09717-0
DOI: 10.32604/cmc.2023.031723

4. Technical Review
Analyzing Methodology and Algorithms:
The methodology is well-structured, incorporating a variety of predictive models and feature selection techniques. However, while the paper presents a good mix of machine learning models, a more detailed explanation of the decision-making process behind choosing specific models (e.g., why Gradient Boosting and XGBoost were selected over other models) would enhance the understanding of the methodology. Additionally, more discussion on the optimization process and the choice of hyperparameters would be beneficial.

Suggestions for Improvement:

Provide a clearer rationale for selecting Gradient Boosting and XGBoost over other potential models.

Include more details about the hyperparameter optimization process and how the settings were selected for the best performance.

Performance Evaluation:
The performance evaluation metrics (RMSE, MAE, R²) are appropriate for the models used. The authors could further strengthen the evaluation by comparing the performance of their models with simpler baseline models, such as linear regression, to highlight the improvements offered by non-linear models. Additionally, including a performance analysis of how the models handle real-time data, such as incorporating LMS interactions in a continuous manner, would add value.

Suggestions for Improvement:

Consider adding comparisons with baseline models like linear regression to demonstrate the advantages of non-linear models.

Expand the performance evaluation by discussing how the models perform when implemented in real-time or with new data.

5. Visualization and Analysis
Checking the Adequacy of Figures, Tables, and Diagrams:
The figures and tables presented in the manuscript are generally well-organized and informative. However, some figures, such as the correlation matrix (Figure 2), could benefit from clearer labels and a more detailed explanation of how the correlations were calculated. The use of multiple histograms to visualize model performance before and after feature selection is a valuable addition but could be accompanied by more explanation regarding the distribution of the autonomous scores.

Suggestions for Improvement:

Ensure that all figures have fully labeled axes and more detailed captions that help the reader understand the significance of each visualization.

Provide more detailed explanations for the visualizations of feature importance and correlation matrices.

Experimental design

.

Validity of the findings

.